complexity/biophysics

collective motion, complex networks, sparse Bayesian learning, Fokker–Planck equation

**Author for correspondence:**
Wei Lin
e-mail: wlin@fudan.edu.cn

# Coordinating directional switches in pigeon flocks: the role of nonlinear interactions

Duxin Chen[1,†], Yongzheng Sun[2,†], Guanbo Shao[1], Wenwu Yu[1], Hai-Tao Zhang[3] and Wei Lin[4,5]

[1]School of Mathematics, Southeast University, Nanjing 211096, People's Republic of China
[2]School of Mathematics, China University of Mining and Technology, Xuzhou 221116, People's Republic of China
[3]School of Artificial Intelligence and Automation, Huazhong University of Science and Technology, Wuhan 430074, People's Republic of China
[4]Research Institute of Intelligent Complex Systems and MOE Frontiers Center for Brain Science, Fudan University, Shanghai 200433, People's Republic of China
[5]School of Mathematical Sciences, LMNS, and SCMS, Fudan University, Shanghai 200433, People's Republic of China

 DC, 0000-0002-3194-2258; YS, 0000-0001-9504-5663;
GS, 0000-0002-4670-9598; WY, 0000-0003-3755-179X;
H-TZ, 0000-0002-8819-8829; WL, 0000-0002-1863-4306

The mechanisms inducing unpredictably directional switches in collective and moving biological entities are largely unclear. Deeply understanding such mechanisms is beneficial to delicate design of biologically inspired devices with particular functions. Here, articulating a framework that integrates data-driven, analytical and numerical methods, we investigate the underlying mechanism governing the coordinated rotational flight of pigeon flocks with unpredictably directional switches. Particularly using the sparse Bayesian learning method, we extract the inter-agent interactional dynamics from the high-resolution GPS data of three pigeon flocks, which reveals that the decision-making process in rotational switching flight performs in a more nonlinear manner than in smooth coordinated flight. To elaborate the principle of this nonlinearity of interactions, we establish a data-driven particle model with two potential wells and estimate the mean switching time of rotational direction. Our model with its analytical and numerical results renders the directional switches of moving biological groups more interpretable and predictable. Actually, an appropriate combination of natures, including high density, stronger nonlinearity in interactions, and moderate strength of noise, can enhance such highly ordered, less frequent switches.

†Duxin Chen and Yongzheng Sun contributed equally to this work.

# 1. Introduction

The underlying mechanism of how collective biological entities move together in the absence of leader(s) or driving field has attracted the attention of both science and engineering communities for a long time [1]. Significant progresses in understanding the nature of these features have been achieved through the development of self-propelled particle (SPP) models [1–3]. These models, where the particles are assumed to orient their velocity parallel to the average velocity in a local neighbourhood, have been used to explain some experimental observations of collective behaviours [4–10]. However, the exact understanding is still unclear largely due to the limitation of data processing capability. As a remedy, fuelled by the advances of data acquisition and data-based learning techniques, empirical studies have shed some light onto deeper understanding of collective motion of biological groups [11–13]. Thereby, other frameworks to understand the interaction mechanisms of collective behaviours have been proposed, e.g. approaches proposed from the viewpoints of correlation analysis [14,15], statistical mechanics [16–18] and complex networks [19,20]. Combining techniques of both data analytics and data-driven modelling likely refines our elaboration of the nature of such complex coordinated systems.

In addition to those homodromous collective motions, unpredictable changes in their direction of movements emerge in biological groups. Examples abound: the whirling switch of marching locusts [21–23], foraging ants [24], fish schools and prawns [25,26], swarming bacteria [27], and the rotational change of bird flocks [28]. Naturally, such systems are reminiscent of two-phase equilibrium systems. Indeed, the seminal model proposed by Vicsek [2] and its variants have been used to explain the directional switching phenomenon of collective entities [21–23,29]. The collective phenomenon of different sizes of bird flocks has already been considered in the context of collective motions [30–36]. However, an in-depth depiction of the interactional mechanism triggering equilibrium transition is still lacking.

Nonlinear interactional mechanism plays important roles in the stability, controllability, and collective dynamics of complex biological and ecological systems [37–40]. For the benefits of theoretical analysis, linear interactions are empirically incorporated into previous models. Due to recent advances in machine learning, the nonlinear interaction mechanism can be identified from empirical data [41–43]. To our best knowledge, in addition to the large population size and presumed linear interactions, there is no result about the significance of nonlinearity in the interactions for achieving highly ordered, directional switches.

In this study, we therefore investigate the directional switches in free flight of pigeon flocks, where individuals hover above their home loft with spontaneous changes of rotational directions, and systematically reveal the interplay between the inter-agent coupling dynamics and the transition of equilibrium states. Based on interactional mechanism revealed through exploiting the real GPS data of pigeon flocks by the sparse Bayesian learning method, this work establishes a data-driven particle model with two potential wells and estimates the mean switching times of rotational direction. We report that a certain degree of nonlinearity in interactions can enhance the coordinated rotational flight of pigeon flocks with infrequent directional switches. Such a mechanism is surprisingly contrary to common knowledge: linear and time-invariant interactions are sufficient to induce ordered collective motions.

# 2. Main results

## 2.1. Extracting the interaction mechanism

To begin with, we focus on the high-resolution GPS data of free flight consisting of more than 30 releases of three pigeon flocks (labelled, respectively, as flocks A, B and C) [31], each of which has 10 individuals and lasts for several minutes. The GPS logger was randomly allocated and affixed to the back of each pigeon, logging time-stamped longitude, latitude and altitude data at 10 Hz. Due to the fact that the average standard deviation of the flight altitude in each release is sufficiently small ($5.22 \pm 1.27$ m), it suffices to only use the $x$-axis and the $y$-axis data for investigation as did previous studies [28,30]. Typical flight trajectories are shown in figure 1a.

Generally, it is often encountered that collective movements of pigeon flocks exhibit an extremely high degree of global ordering. This indicates that the system reaches a dynamical equilibrium with no macroscopic change of the intensity of speeds. To sensitively sharpen and extract the tiny changes of each individual, we analogously introduce a concept of internal angular momentum from quantum

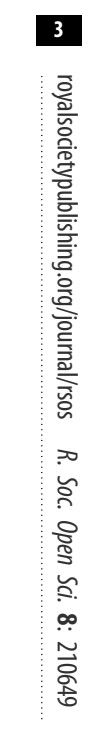

**Figure 1.** Smooth and switching flights of pigeon flocks. (*a*) Typical flight trajectories in different planes. (*b*) Dynamics of the normalized spins of three randomly selected individuals in flock A. (*c*) Distribution of the number of neighbours. (*d*) Distribution of the order of interactions. Here, the linear interactions are dominant in the smooth part, while the percentage of high-order interactions grows larger in switching trajectories.

mechanics [44], namely individual spin, $\vec{s}_i = \chi_i \cdot (\vec{v}_i \times (d\vec{v}_i/dt))$. Here, $\chi_i$ and $v_i$, respectively, denote the $i$th individual's moment of inertia and its velocity. For simplicity, consider every turning as a part of uniform circular motion, and neglect the inertia term, i.e. $\vec{s}_i = \vec{v}_i \times (d\vec{v}_i/dt)$. Thus, the index $\vec{s}_i$ is orthogonal to both the velocity and the differential term of velocity, and its positive and negative normalized values stand, respectively, for the clockwise (CW) and the counterclockwise (CCW) rotations through simply using the right-hand screw rule. Suppose the free flight of pigeon flocks evolving in a two-dimensional plane, and also suppose $\vec{s}_i$ to be a one-dimensional index that is simplified as $s_i$ afterwards. Denote by $S_a(t) = 1/N \sum_{i=1}^{N} \text{sgn}[s_i(t)]$ an order parameter, where $\text{sgn}[\cdot]$ denotes the signum function. This order parameter characterizes the collective decision made by the entire flock. As usual, the zero value of $S_a$ corresponds to reciprocally conflicted states with half supporters and half opponents, while $S_a(t) = \pm 1$ corresponds to consensus of decision-making, respectively, in CW and CCW rotations. Here, in the entire dataset, more than 96% of states are of $S_a(t) = \pm 1$ during the collective flight.

In figure 1*b*, we depict the dynamics of the normalized $s_i$ for three individuals in flock A. The fluctuations of $s_i$ indicate the willing of one individual to perform the movement, and the symbol changes of $s_i$ reflect the switch of rotational direction in the free flight. With the influence of the individual's neighbours, the dynamics show a higher degree of order in collective rotations. Since $s_i$ encrypts the self-dynamics and the interaction mechanism with others, we next seek to reveal the inter-agent interactions through investigating the evolving dynamics $s_i$. For this purpose, a sparse Bayesian learning (SBL) method [45] is used to extract the inter-agent interactions among the individuals. The method employs strictly dynamical program instructions by building up a model from input data and thereby making data-driven prediction of the future evolution of the entire flock. Essentially, the method relies on an assumption that it is possible to capture the system dynamics by

designing a finite candidate set of dictionary functions that are obtained from prior knowledge. Meanwhile, it adopts prior probabilities to represent system uncertainty via both probabilistic rules and inferential processes.

Concretely, we integrate the data of three flocks with the SBL formulation which reads

$$Y = \Phi\Omega + \xi,$$ (2.1)

where $Y$ is the collected time-series data denoted in a state-space form, $\Phi$ is the dictionary function matrix which gathers prior knowledge of the input data with potential over-complete formulations, $\Omega$ is the coefficient matrix including the objective connection information, and $\xi$, the addictive process noise during the circular flights, is assumed to be independent and identically distributed Gaussian with zero mean. Set $Y = [s_i(t_1), s_i(t_2), \ldots, s_i(t_q)]^T$ as the $q$-step input data of $s_i(t)$. Inspired by the approximating ability of Taylor's expansion, we adopt the polynomials as the elements of the dictionary matrix in (2.1):

$$\Phi = \begin{bmatrix} f_{N_i}(t_0) & f_{N_i}(t_0)^2 & \cdots & f_{N_i}(t_0)^5 \\ f_{N_i}(t_1) & f_{N_i}(t_1)^2 & \cdots & f_{N_i}(t_1)^5 \\ \vdots & \vdots & \ddots & \vdots \\ f_{N_i}(t_{q-1}) & f_{N_i}(t_0)^2 & \cdots & f_{N_i}(t_{q-1})^5 \end{bmatrix}$$ (2.2)

to cover the coupling dynamics as much as possible. Here, the function $f_{N_i}(t)^k = [(s_{j_1}(t) - s_i(t))^k, (s_{j_2}(t) - si(t))k, \ldots, (s_{j_{N-1}}(t) - s_i(t))^k]$ with $k = 1, \ldots, 5$ and $N_i$, the neighbourhood of individual $i$. To reduce the computational cost, we ignore the influence from the orders higher than 5 and have the coefficient matrix as

$$\Omega = [\underbrace{\cdots}_{\omega_1} \ \underbrace{\cdots}_{\omega_2} \ \underbrace{\cdots}_{\omega_3} \ \underbrace{\cdots}_{\omega_4} \ \underbrace{\cdots}_{\omega_5}]^T,$$

where $\omega_i$ denotes the sub-matrix that consists of the coefficients related to the $i$th order term in $\Phi$. Therein, $f_{N_i}(t)$ includes the position difference between individual $i$ and all the others ($j \neq i$). The coefficient matrix contains the connectivity information between individual $i$ and the others.

Now, we employ the SBL method formulated in (2.1) to every switching segment (5 s before and after a successful rotational direction change) and smooth segment (except for the switching segment) among the entire input data of 30 releases of three pigeon flocks. We solve the matrix multiplication (convex) optimization problem through a procedure relying on an efficient iterative re-weighted $\ell_1$-minimization algorithm [45,46]. With a higher value of $q$, i.e. the length of input data, the estimation error of SBL grows higher. Throughout the computation, we set $q = 5$ with an absolute estimation error less than 0.5(Nms). If the individual's trajectory to be recovered is influenced by the others indicated by the coefficient matrix, it is considered to interact with a focal one. For more details of the SBL set-up for identifying the interactional dynamics of pigeon flocks, refer to [28] and the electronic supplementary material.

Previous studies [17,47] suggested that the number of neighbours was around 6–7 in starling flocks. Analogously, we collect the degree distribution and show it in figure 1$c$. Clearly, for our data, the number of neighbours of an individual mostly concentrates between 2 and 4. Such a local interaction feature is reminiscent of the fixed number of neighbours in large-sized starling flocks [17]. The entire free-flight trajectories can be generally divided into two types: The smooth trajectories and the switching ones. While the former ones show a much higher degree of coordination, the latter ones encipher a complicated decision-making process. As shown in figure 1$d$, the distributions of the two types of dynamics, revealed by the SBL method, are majorly different. Linear interactions (synchronous changes of state) among individuals are dominant in the smooth part of free flight; however, the percentage of high-order (orders higher than 2 identified by the SBL method) interactions grows higher along the switching trajectories. These indicate the interactions flourishing in nonlinear or even more complicated style, so that decision-making likely induces a high degree of inter-agent interactions of pigeon flocks.

## 2.2. Model of coordinating directional switches

By considering an ordered switching of rotational directions and disregarding spatial position and velocity of the individuals, we consider a model in the Vicsek style [2], where the interactions are mediated by the spins. Also, we assume that each individual adjusts its behaviour according to its

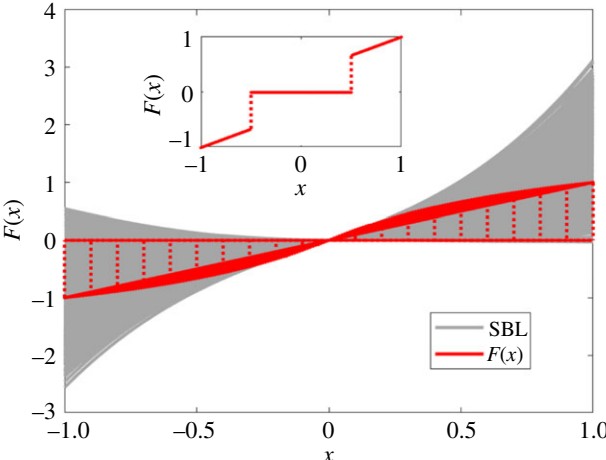

**Figure 2.** Interaction function: from linear form to nonlinear form. Here, the functions (grey shadow) identified using the SBL method display switching nature around $x = 0$, while the function $F$, defined in (2.4) with $\alpha$ away from 0 and depicted also in the inset, reflects such nonlinear nature. Here, $\alpha \in (0, 1)$ tunes the linearity or nonlinearity intensity of $F$ specified in (2.4).

neighbours whose index set is defined by $J_i(t) = \{j \in \{1, \ldots, N\} \mid a_{ij} > 0\}$. An individual's behaviour is described by its spin, $s_i = s_i(t)$ with $i = 1, \ldots, N$, which obeys the following dynamics:

$$ds_i = \left[ F(S_i^{\text{local}}(t)) - s_i(t) \right] dt + \eta \, dW_i, \tag{2.3}$$

where $\eta > 0$ quantifies the noise strength, each $dW_i$, representing the environmental perturbation, is set as the standard white noise (independently sampled for each individual), and $S_i^{\text{local}}(t) = (1/|J_i(t)|) \sum_{j \in J_i(t)} s_j(t)$ is the average spin at time $t$ of the individuals within the local neighbourhood of the $i$th individual. Denote by $S(t)$ the average spin of the whole group. So, when the networks have a complete interacting graph, $S(t) \equiv S_i^{\text{local}}(t)$ for arbitrary $i$. More importantly, inspired by the complex interacting patterns that are discovered based on the empirical data (figure 1), we design a one-parameter function combining its argument and the sign of this argument as

$$F(x) = \begin{cases} \omega_\alpha x + (1 - \omega_\alpha) \text{sgn}(x), & |x| > \alpha, \\ 0, & |x| \le \alpha, \end{cases} \tag{2.4}$$

where $\omega_\alpha = 1/(1 + \alpha)$ and $\alpha \in [0, 1]$ is an adjustable parameter, determining the relative weight in a decision-making manner that individuals balance the tradeoff between their spin direction and the alignment to their neighbours. Actually, as shown in figure 2, the interaction functions revealed by the SBL method display switching nature around $x = 0$. Also shown in figure 2, the function $F$ can majorly reflect such nonlinear nature as $\alpha$ is away from 0. Particularly, $F$ vanishes in between $[-\alpha, \alpha]$, so that the interaction among the individuals is determined by the diffusion term in equation (2.3), whereas, outside the interval, $F$ takes its value dependent on the averaged neighbourhood spin. In addition, $F$ becomes linear as $\alpha$ is at or very close to 0. Hence, both linear and nonlinear interactions are encrypted in the selection of the parameter $\alpha$ for the piecewise function $F$.

Next, we numerically investigate how $\eta$, the intensity of noise, $\alpha$, the nonlinear extent of interactions, and $N$, the group number, separately or jointly impact the switching of rotational direction. Each result is obtained through averaging over 100 realizations of 20 000-step (2000 s) update in the simulation with different selections of the parameters $\eta$ and $\alpha$. To focus on the interactional dynamics, we neglect the position influence of neighbour selection in the present spin model (2.3). The number of neighbours in all the simulations is randomly selected from the set {2, 3, 4} in a group of ten individuals because of the distribution displayed in figure 1c using the SBL method. We depict the results of the mean number of switches in figure 3, showing that a moderate strength of noise may induce an ordered switch of rotational direction, and that the mean number of switches increases as $\alpha$ becomes smaller. Therefore, for a given strength of noise, the larger the value of $\alpha$ (i.e. the stronger the nonlinearity in interactions), the less frequently the switches emerge in the rotational direction for pigeon flocks. Interestingly, the role of a stronger nonlinearity in the interactions is somewhat akin to the influence of a larger $N$, the population size.

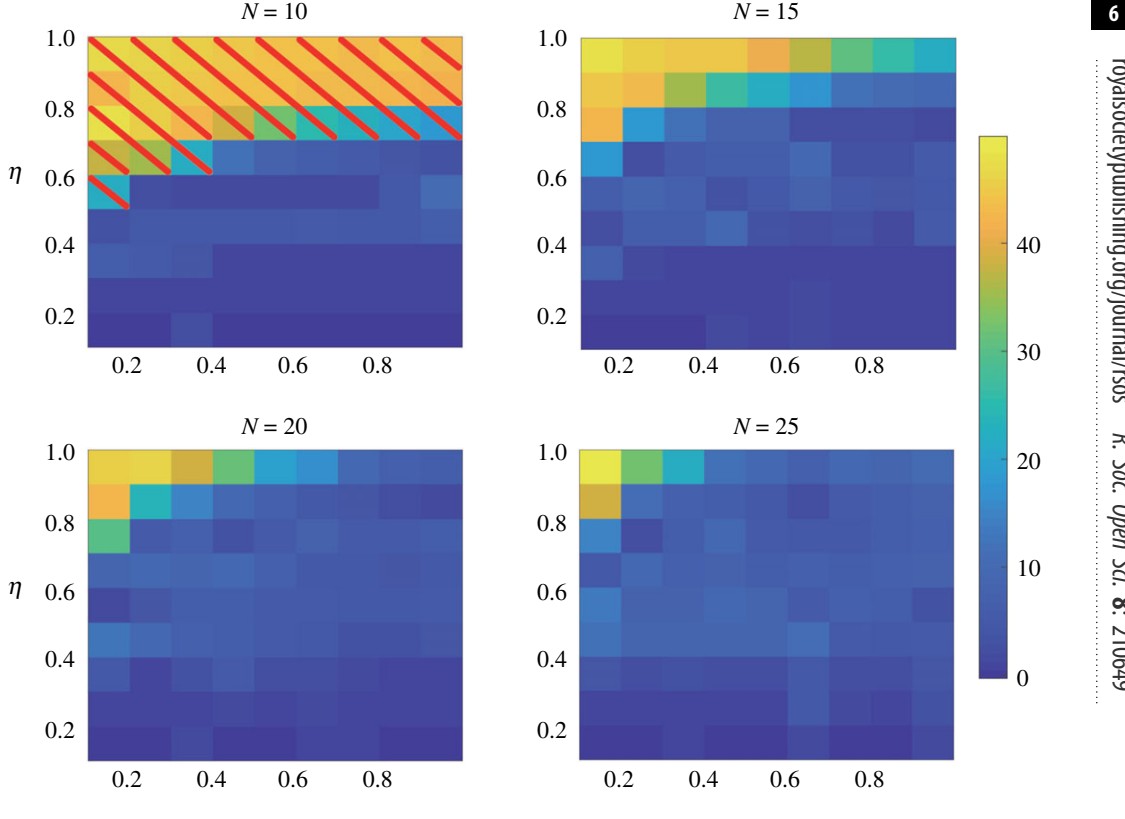

**Figure 3.** Mean number of switches of groups with local interactions. Here, with different selection of $(\alpha, \eta)$ and $N$ for model (2.3) using local interactions, the colour represents the mean number of switches, and the number of neighbours is randomly selected from the set $\{2, 3, 4\}$ as discovered from the pigeon flocks data with $N = 10$. The red shadow in the first panel corresponds to the empirical result of the flocks with 10 pigeons.

## 2.3. Calculation of the mean first passage time

To deepen our understanding of the mechanisms behind the generation of ordered motion of the SPP model, we turn to analyse the dynamics (2.3), in which all individuals are supposed to be interacting with each other. The advantage of such configurations can make it possible to obtain analytical results for this model without losing the elementary nature of switches in flocks. As such, the local average spin $S_i^{\text{local}}$ in equation (2.3) becomes the global average $S(t)$ as mentioned above. Thus, equation (2.3) becomes $dS_i = [F(S(t)) - S_i(t)]\,dt + \eta\,dW_i$, which, according to the theory established in [48], yields the following Itô stochastic differential equation for $S(t)$:

$$dS = \Big[F(S(t)) - S(t)\Big]\,dt + \eta N^{-1/2}\,dW. \tag{2.5}$$

Denote by $P(s, t)$ the probability density of the stochastic process produced by equation (2.5). Then, $P(s, t)\,ds$ is the probability of the global average spin $S(t) \in [s,\ s + ds)$ and $P(s, t)$ satisfies the Fokker–Planck equation [49] as follows:

$$\frac{\partial}{\partial t}P(s, t) = \frac{\partial^2}{\partial s^2}\left[\frac{\eta^2}{2N}P(s, t)\right] - \frac{\partial}{\partial s}[(F(s) - s)P(s, t)]. \tag{2.6}$$

Denote by $P_{\text{st}}(s) = \lim_{t \to \infty} P(s, t)$ the stationary probability distribution of $S(t)$. We thus obtain from (2.6) that

$$\frac{d^2}{ds^2}\left[\frac{\eta^2}{2N}P_{\text{st}}(s)\right] - \frac{d}{ds}[(F(s) - s)P_{\text{st}}(s)] = 0,$$

which further implies $P_{\text{st}}(s) = (CN/\eta^2)\exp[-\phi(s)]$, where $C$ is a normalization constant satisfying $\int_{-\infty}^{\infty} P_{\text{st}}(s)\,ds = 1$ and the potential $\phi(s) = (N(1 - \omega_\alpha)/\eta^2)(s^2 - 2|s|) + (N/\eta^2)[\omega_\alpha \alpha^2 + 2(1 - \omega_\alpha)\alpha]$ when

$|s| > \alpha$, and $\phi(s) = (Ns^2/\eta^2)$ when $|s| \leq \alpha$. Therefore, we obtain

$$P_{st}(s) = \begin{cases} \frac{CN}{\eta^2} e^{-(Ns^2/\eta^2)}, & |s| \leq \alpha, \\ \frac{C'N}{\eta^2} e^{-(N(1-\omega_\alpha)/\eta^2)(s^2-2|s|)}, & |s| > \alpha, \end{cases} \tag{2.7}$$

where we set, respectively, $C = (\eta^2/2N(L + \alpha - L\alpha))$ and $C' = LC$, where $L = \exp[(N/\eta^2)[\omega_\alpha\alpha^2 + 2(1 - \omega_\alpha)\alpha]]$. Clearly, $P_{st}(s)$ in (2.7) has two local maxima at $s = \pm 1$, which reinforces a real phenomenon where the equilibria that individuals fly coordinately along one direction occupy more than 96% during the collective flight of the entire flock.

We further calculate $T(s_\pm \rightarrow s_\mp)$, the *mean first passage time* from one well (equilibrium) of the effective potential to the other. Assume $S = -1$ at $t = 0$, and $\tau$ is the first time that the group spin $S$ escapes from the interval $(-\infty, 0]$. Then, $\tau$ satisfies $[F(s) - s](d\tau/ds) + (\eta^2/2N)(d\tau^2/ds^2) = -1$, which immediately yields

$$\tau = \frac{2N}{\eta^2} \int_{-1}^{0} \frac{1}{P_{st}(s)} \, ds \int_{-\infty}^{s} P_{st}(\xi) \, d\xi. \tag{2.8}$$

Introducing $\mathrm{erf}(x) = (2/\sqrt{\pi}) \int_0^x e^{-t^2} \, dt$, the *Gauss error function*, and $\mathrm{erfi}(x) = (2/\sqrt{\pi}) \int_0^x e^{t^2} \, dt$, the *imaginary error function*, we obtain the tipping point that the group spin $S$ escapes from one well of the potential to the other as

$$\tau \simeq \mathcal{T}_{\alpha,N} \cdot \exp\left[\frac{N(1 - \omega_\alpha)(1 + 3\alpha)}{\eta^2}\right] \tag{2.9}$$

with $\mathcal{T}_{\alpha,N} = \sqrt{\pi/(1 - \omega_\alpha)}[\sqrt{\pi} + \mathrm{erf}(\sqrt{N(1 - \omega_\alpha)}/\eta)]\mathrm{erfi}(\sqrt{N/\eta^2})$ (refer to electronic supplementary material for detailed calculations). In figure 4, we compare the value of $\tau$ estimated in (2.9) with the numerical result for differently selected $\alpha$, $\eta$ and $N$. Evidently, the analytically estimated mean first passage time agrees well with the simulation results: the stronger nonlinearity in interactions with a certain strength of noise means that switches emerge less frequently.

## 3. Discussion

Understanding the coordination mechanism of bird flocks has been a long-standing open problem in communities of science and engineering. By using the SBL method, the interaction rules governing highly coordinated flight of pigeon flocks have been analysed in our previous work [28]. However, the results obtained in [28] are based on a linear interaction assumption motivated by the standard Vicsek model. In this work, different from [28], we extended the usage of the SBL method and extracted the linearity or/and nonlinearity of the interaction from the smooth and the transient switching trajectories. We focus on the influence of nonlinear interaction on the directional switches of pigeon flocks.

To describe the two equilibria (i.e. the cooperative CW and CCW rotations) in the free flight of pigeon flocks, we have established the symmetrical double-potential-well SPP model with both linear and nonlinear interactions among the individuals. The SPP model is reminiscent of Arnold's tongues in synchronization theory [50], where the strength of coupling has a strong influence on the locking region of synchronization. Analogously, the parameter $\alpha$ in our stochastic differential model balances the linearity and nonlinearity of coupling strength, which influences the number of switches. Based on the theory of the Fokker–Planck equation, we have obtained the analytical estimations of the mean switching time of directional switches which allow us to evaluate the influence of noise strength and the nonlinearity extent of interaction on the coordination of pigeon flocks. We have shown that, appropriately tuning the nonlinearity extent of interaction and the noise strength for a given group of entities, the phenomena of random switches of rotational direction can be commendably reproduced to a high fidelity. The error exists when we are calculating the value of $P_{st}$. Since the function can be approximately evaluated as the normal distribution function, we have shown that if $0 < \sqrt{2N(1-\omega)/\eta^2}(\alpha - 1) < 0.8$, the global error is no larger than 7.25%. Refer to Methods for detailed error analysis.

We, therefore, believe that our findings as well as our model can provide some insight onto a more in-depth understanding of highly coordinated collective motion of animal groups with spontaneous changes of the equilibria and thus are beneficial to constructing biologically inspired, large-scale devices in the near future.

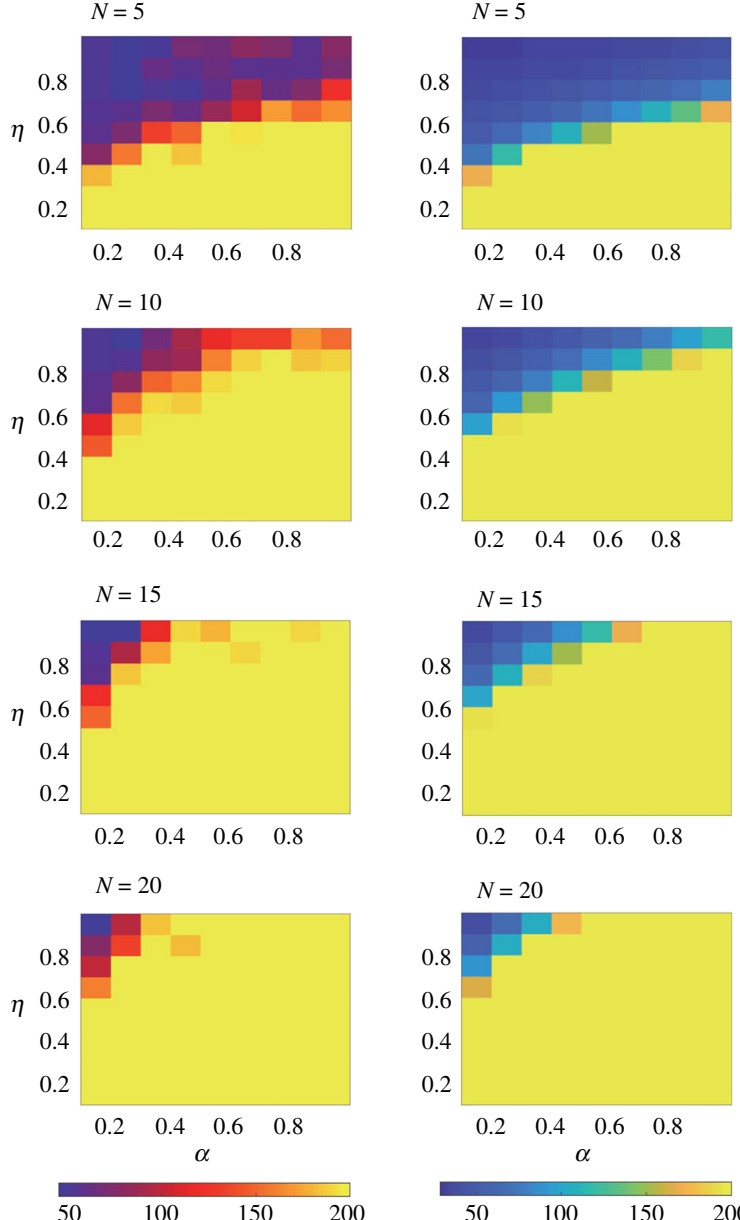

**Figure 4.** The mean first passage time for the pigeon flocks. Here, the colour stands for the mean first passage time, and provided are the comparisons of the numerical results (left) with the analytically estimated results (right), under different selections of the parameters $\alpha$ and $\eta$ for given size $N$. The corresponding results are highly consistent.

# 4. Methods

## 4.1. Sparse Bayesian learning algorithm

Bayesian theory, as a branch of statistics, makes full use of prior knowledge. The core idea of SBL is to assume that all unknown parameters are random variables that match a certain probability distribution. In practical problems, such as signal transmission and image processing, we expect that the final result of processing the data is a sparse solution, so that the key potential relationship inside the data can be quickly extracted. In general, when the dictionary matrix $\Phi$ in SBL is a positive definite kernel function, the correlation vector machine algorithm is obtained. In this study, we use an $l_1$-regularized SBL algorithm to identify nonlinear discrete time series with noise. The key point is to add an $l_1$-norm to the cost function, so that the obtained result satisfies sparsity. Refer to electronic supplementary material for more detailed illustration.

## 4.2. Error analysis

The error exists when we are calculating the value of $P_{st}$. Refer to electronic supplementary material for detailed calculation of $P_{st}$. Let $\Phi(x)$ represent the distribution function of the standard normal distribution, i.e.

$$\Phi(x) = \frac{1}{\sqrt{2\pi}} \int_{-\infty}^{x} \exp\left(-\frac{t^2}{2}\right) dt. \tag{4.1}$$

Note that the Taylor expansion of $\Phi(x)$ around $x = 0$ can be written as

$$\Phi(x) \approx \Phi(0) + \Phi'(0)x = \frac{1}{2} + \frac{1}{\sqrt{2\pi}} x. \tag{4.2}$$

Since $P_{st}$ can be approximately evaluated by the normal distribution function, we thus focus on the relative error function (REF), which can be given as

$$\text{REF} = \frac{\{\Phi(M) - \Phi(-M)\} - \{(2M/\sqrt{2\pi})\}}{\Phi(M)}. \tag{4.3}$$

Here, $M$ satisfies

$$M = \sqrt{\frac{2N(1-\omega)}{\eta^2}} (1 - \alpha). \tag{4.4}$$

Then,

$$\begin{aligned} \text{REF} &= \left| \frac{\{\Phi(M) - \Phi(-M)\} - \{(2M/\sqrt{2\pi})\}}{\Phi(M)} \right| \\ &= \left| \frac{\{2\Phi(M) - 1\} - \{(2M/\sqrt{2\pi})\}}{\Phi(M)} \right|. \end{aligned} \tag{4.5}$$

Next, we show some properties of $\text{REF}_\Phi$ as follows. First,

$$\begin{aligned} \frac{\partial \text{REF}_\Phi}{\partial M} &= \frac{((2/\sqrt{2\pi})\exp-(M^2/2) - (2/\sqrt{2\pi}))(1/\sqrt{2\pi})\int_{-\infty}^{M}\exp(-(t^2/2))dt}{\Phi(M)} \\ &\quad - \frac{[2(1/\sqrt{2\pi})\int_{-\infty}^{x}\exp(-(t^2/2))dt - 1 - (2M/\sqrt{2\pi})](2/\sqrt{2\pi})\exp(-(M^2/2))}{\Phi(M)}. \end{aligned} \tag{4.6}$$

By direct calculation, we obtain that $\partial \text{REF}_\Phi / \partial M$ is always positive if $M > 0$. Thus, we obtain that $\text{REF}_\Phi(x_2) > \text{REF}_\Phi(x_1)$ for $x_2 > x_1$. This implies that $\text{REF}_\Phi$ is monotonically increasing. Then, it is straightforward to get the evaluation of $\text{REF}_\Phi$ as follows:

$$\left.\begin{aligned} \text{REF}_\Phi(0) &= 0, \\ \text{REF}_\Phi(0.8) &\approx 7.81\% \\ \text{REF}_\Phi(0.9) &\approx 11.03\%. \end{aligned}\right\} \tag{4.7}$$

and

Additionally, we investigate the change of REF with respect to $C$, denoted by $\text{REF}_C$, as follows:

$$\text{REF}_C(M) \approx 1 - \frac{1}{1 + \text{REF}_\Phi}, \tag{4.8}$$

with

$$\left.\begin{aligned} \text{REF}_C(0) &= 0, \\ \text{REF}_C(0.8) &\approx 7.25\% \\ \text{REF}_C(0.9) &\approx 13.05\%. \end{aligned}\right\} \tag{4.9}$$

and

Thus, if $0 < \sqrt{2N(1-\omega)/\eta^2}(\alpha - 1) < 0.8$, the global error is no larger than $7.25\%$.

Data accessibility. Data and relevant code for this research work are stored in GitHub: https://github.com/cccs-data/pigeon_spin and have been archived within the Zenodo repository: https://doi.org/10.5281/zenodo.5383147. All relevant computer codes are available from the authors upon request.

Authors' contributions. D.C., Y.S., W.Y. and W.L. conceived the project, D.C., Y.S., W.Y., H.T.Z. and W.L. analysed the model, D.C., Y.S. and G.S. performed the experiments, all analysed the data. W.L. wrote the paper with help from D.C. and Y.S.

Competing interests. We declare we have no competing interests.

Funding. This work was supported by the Fundamental Research Funds for the Central Universities (no. 2242019K40111), the National Natural Science Foundation of China (grant nos. 61903079, 62073076, 61403393, 61391240193, U1713203, 51729501 and 11925103), the Natural Science Foundation of Jiangsu Provience (grant no. BK20211241) and Hubei Province (grant no. 2019CFA005),  and partially by the STCSM (grant nos. 2021SHZDZX0103, 19511101404 and 19511132000).

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
