## [Peer Review File · Royal Society Open Science]

Review History

RSOS-210649.R0 (Original submission)

Review form: Reviewer 1

Is the manuscript scientifically sound in its present form?

Yes

Are the interpretations and conclusions justified by the results?

Yes

Is the language acceptable?

Yes

Do you have any ethical concerns with this paper?

No

Have you any concerns about statistical analyses in this paper?

No

Recommendation?

Accept as is

Comments to the Author(s)

Long range ordered flocking is one of the most interesting feature of active matter systems, exhibiting non-equilibrium structures without being driven. This work addresses the problem of directional switches in coordinated complex systems, taking pigeon flocks data for their data-based modelling, backed up by careful statistical analysis. The aim is to understand the mechanism leading to transition due to the interactions in the population, which plays a key role in the collective dynamics of ecological systems. Previous models concentrate on linear interaction, this work extracts the significance of nonlinearity to achieve the ordered directional switches. The interaction mechanism is obtained by using sparse Bayesian learning method on GPS data of pigeon flocks. The work is in fact, a data-driven particle model on a double potential well driving the switching dynamics. Their mathematical analysis of the mean first passage time is sufficiently rigorous to validate the data-driven modelling.

The introduction, data analysis, data-based modelling, discussion of results, and looking forward are clearly stated and easy to read. In fact, it was a pleasure to read this manuscript.

I recommend its publication the the J. of the Royal Society Interface as is.

Review form: Reviewer 2 (Eckehard Schöll)**Is the manuscript scientifically sound in its present form?**

Yes

Are the interpretations and conclusions justified by the results?

Yes

Is the language acceptable?

Yes

Do you have any ethical concerns with this paper?

No

Have you any concerns about statistical analyses in this paper?

No

Recommendation?

Accept with minor revision (please list in comments)

Comments to the Author(s)

Referee Report on RSOS-210649 by Chen, Duxin et al.

In this paper the role of nonlinear interactions in the coordination of directional switches in pigeon flocks is investigated. The authors present results based upon data-driven, analytical, and numerical methods. First, using the sparse Bayesian learning method, they extract the inter-agent coupling dynamics from GPS data of three pigeon flocks. Then, they present a variant of the Vicsek model of self-propelled particles with a particular nonlinear interaction between the agents (pigeons). Finally they estimate the mean switching time of rotational direction using a Fokker-Planck equation by calculating the first-passage time between two potential wells representing the two equilibria, i.e., clockwise and counter-clockwise rotation.

This is an interesting topic, the results are sound, and the paper is well written. The new findings are the nonlinear interactions of inter-agent coupling dynamics which is modeled by Eq.(4). This adds to the understanding of coordinated rotational flight of pigeon flocks with infrequent directional switches.

However, before I can recommend publication in Royal Society Open Science, the authors should take care of the following issues:

(i) There seems to be a large overlap with a previous paper of some of the authors [28]. In the Introduction and in the Results Section it should be clearly stated in which respects this paper differs from Ref.[28] Chen D, Xu B, Zhu T, Zhang HT. 2017 Anisotropic interaction rules in circular motions of pigeon flocks: An empirical study based on sparse bayesian learning.

(ii) All quantities should be defined when they first occur; e.g., on p.2 left, line 54, v_i (velocity, I assume) is not defined, and the formula for spin is not explained sufficiently (as angular momentum I would expect $r \times p$). It is not clear why this is a type of Vicsek model.

(iii) I assume that x in Eq. (4) and in Fig.2 is not the position coordinate of Fig.1a, as in Ref. [28], but the average spin S . Please clarify.

(iv) The nonlinear interaction function Eq.(4) is reminiscent of Arnold tongues in synchronization theory. Could the authors comment on that? The chosen value of α might then be compared to an effective coupling strength.

Minor remarks:

On p.2 right, line 30: give a reference for sparse Bayesian learning method

p.2 right, line 50: define i.i.d.

p.4 right, line 51, p.5 left, line 36 + 55 etc.: equilibria NOT equilibriums

p.5 left, line 23-25: sentence is not comprehensible

p.5 left, line 41: shown

Fig.2: The caption should better describe the role of α in the different red curves displayed.

Give the range and the values of α .

Bibliography: - give all names of coauthors

- error in author names in Ref. 29

Decision letter (RSOS-210649.R0)

Dear Dr Lin

On behalf of the Editors, we are pleased to inform you that your Manuscript RSOS-210649 "Coordinating directional switches in pigeon flocks: The role of nonlinear interactions" has been accepted for publication in Royal Society Open Science subject to minor revision in accordance with the referees' reports. Please find the referees' comments along with any feedback from the Editors below my signature.

Please submit your revised manuscript and required files (see below) no later than 7 days from today's (ie 24-Aug-2021) date. Note: the ScholarOne system will 'lock' if submission of the revision is attempted 7 or more days after the deadline. If you do not think you will be able to meet this deadline please contact the editorial office immediately.

on behalf of Professor Roland Bouffanais (Associate Editor) and Pietro Cicuta (Subject Editor)
openscience@royalsociety.org

Associate Editor Comments to Author (Professor Roland Bouffanais):

Associate Editor: 1

Comments to the Author:

We are kindly asking you to make another round of revision to address some minor comments made by one reviewer.

Reviewer comments to Author:

Reviewer: 1

Comments to the Author(s)

Long range ordered flocking is one of the most interesting feature of active matter systems, exhibiting non-equilibrium structures without being driven. This work addresses the problem of directional switches in coordinated complex systems, taking pigeon flocks data for their data-based modelling, backed up by careful statistical analysis. The aim is to understand the mechanism leading to transition due to the interactions in the population, which plays a key role in the collective dynamics of ecological systems. Previous models concentrate on linear interaction, this work extracts the significance of nonlinearity to achieve the ordered directional switches. The interaction mechanism is obtained by using sparse Bayesian learning method on GPS data of pigeon flocks. The work is in fact, a data-driven particle model on a double potential well driving the switching dynamics. Their mathematical analysis of the mean first passage time is sufficiently rigorous to validate the data-driven modelling.

The introduction, data analysis, data-based modelling, discussion of results, and looking forward are clearly stated and easy to read. In fact, it was a pleasure to read this manuscript.

I recommend its publication the the J. of the Royal Society Interface as is.

Reviewer: 2

Comments to the Author(s)

Referee Report on RSOS-210649 by Chen, Duxin et al.

In this paper the role of nonlinear interactions in the coordination of directional switches in pigeon flocks is investigated. The authors present results based upon data-driven, analytical, and numerical methods. First, using the sparse Bayesian learning method, they extract the inter-agent coupling dynamics from GPS data of three pigeon flocks. Then, they present a variant of the Vicsek model of self-propelled particles with a particular nonlinear interaction between the agents (pigeons). Finally they estimate the mean switching time of rotational direction using a Fokker-Planck equation by calculating the first-passage time between two potential wells representing the two equilibria, i.e., clockwise and counter-clockwise rotation.

This is an interesting topic, the results are sound, and the paper is well written. The new findings are the nonlinear interactions of inter-agent coupling dynamics which is modeled by Eq.(4). This adds to the understanding of coordinated rotational flight of pigeon flocks with infrequent directional switches.

However, before I can recommend publication in Royal Society Open Science, the authors should take care of the following issues:

- (i) There seems to be a large overlap with a previous paper of some of the authors [28]. In the Introduction and in the Results Section it should be clearly stated in which respects this paper differs from Ref.[28] Chen D, Xu B, Zhu T, Zhang HT. 2017 Anisotropic interaction rules in circular motions of pigeon flocks: An empirical study based on sparse bayesian learning.
- (ii) All quantities should be defined when they first occur; e.g., on p.2 left, line 54, v_i (velocity, I assume) is not defined, and the formula for spin is not explained sufficiently (as angular momentum I would expect $r \times p$). It is not clear why this is a type of Vicsek model.
- (iii) I assume that x in Eq. (4) and in Fig.2 is not the position coordinate of Fig.1a, as in Ref. [28], but the average spin S . Please clarify.
- (iv) The nonlinear interaction function Eq.(4) is reminiscent of Arnold tongues in synchronization theory. Could the authors comment on that? The chosen value of α might then be compared to an effective coupling strength.

Minor remarks:

On p.2 right, line 30: give a reference for sparse Bayesian learning method

p.2 right, line 50: define i.i.d.

p.4 right, line 51, p.5 left, line 36 + 55 etc.: equilibria NOT equilibriums

p.5 left, line 23-25: sentence is not comprehensible

p.5 left, line 41: shown

Fig.2: The caption should better describe the role of α in the different red curves displayed. Give the range and the values of α .

Bibliography: - give all names of coauthors

- error in author names in Ref. 29

===PREPARING YOUR MANUSCRIPT===

===PREPARING YOUR REVISION IN SCHOLARONE===

<https://royalsociety.org/journals/authors/author-guidelines/#supplementary-material> to include a suitable title and informative caption. An example of appropriate titling and captioning may be found at https://figshare.com/articles/Table_S2_from_Is_there_a_trade-off_between_peak_performance_and_performance_breadth_across_temperatures_for_aerobic_scooping_in_teleost_fishes_/3843624.

Author's Response to Decision Letter for (RSOS-210649.R0)

See Appendix A.

Decision letter (RSOS-210649.R1)

Dear Dr Lin,

I am pleased to inform you that your manuscript entitled "Coordinating directional switches in pigeon flocks: The role of nonlinear interactions" is now accepted for publication in Royal Society Open Science.

on behalf of Professor Roland Bouffanais (Associate Editor) and Pietro Cicuta (Subject Editor)
openscience@royalsociety.org

Appendix A

Reponse to the reviewers' comments on the manuscript RSOS-210649

Journal Name: *Royal Society Open Science*

Manuscript ID: *RSOS-210649*

Manuscript Title: *Coordinating directional switches in pigeon flocks: The role of nonlinear interactions*

Authors: *Duxin Chen, Yongzheng Sun, Guanbo Shao, Wenwu Yu, Hai-Tao Zhang and Wei Lin*

Dear Editor and Reviewers,

Thanks for your efforts on handling our manuscript, and the constructive comments and suggestions, which are very useful for us to improve the manuscript. We have revised the manuscript accordingly with the modifications marked in blue. The detailed responses are listed point by point below:

Associate Editor:

General Comment: *We are kindly asking you to make another round of revision to address some minor comments made by one reviewer.*

► Thanks very much for your efforts on handling our manuscript and the comment. We have carefully revised the manuscript according to the valuable review comments. Please refer to the revised version for more details.

Response to Reviewer 1:

General Comment: *Long range ordered flocking is one of the most interesting feature of active matter systems, exhibiting non-equilibrium structures without being driven. This work addresses the problem of directional switches in coordinated complex systems, taking pigeon flocks data for their data-based modelling, backed up by careful statistical analysis. The aim is to understand the mechanism leading to transition due to the interactions in the population, which plays a key role in the collective dynamics of ecological systems. Previous models concentrate on linear interaction, this work extracts the significance of nonlinearity to achieve the ordered directional switches. The interaction mechanism is obtained by using sparse Bayesian learning method on GPS data of pigeon flocks. The work is in fact, a data-driven particle model on a double potential well driving the switching dynamics. Their mathematical analysis of the mean first passage time is sufficiently rigorous to validate the data-driven modelling.*

The introduction, data analysis, data-based modelling, discussion of results, and looking forward are clearly stated and easy to read. In fact, it was a pleasure to read this manuscript.

I recommend its publication the the J. of the Royal Society Interface as is.

► Thanks very much for your careful review and positive recommendation on our manuscript.

Response to Reviewer 2:

General Comment: *In this paper the role of nonlinear interactions in the coordination of directional switches in pigeon flocks is investigated. The authors present results based upon data-driven, analytical, and numerical methods. First, using the sparse Bayesian learning method, they extract the inter-agent coupling dynamics from GPS data of three pigeon flocks. Then, they present a variant of the Vicsek model of self-propelled particles with a particular nonlinear interaction between the agents (pigeons). Finally they estimate the mean switching time of rotational direction using a Fokker-Planck equation by calculating the first-passage time between two potential wells representing the two equilibria, i.e., clockwise and counter-clockwise rotation.*

This is an interesting topic, the results are sound, and the paper is well written. The new findings are the nonlinear interactions of inter-agent coupling dynamics which is modeled by Eq.(4). This adds to the understanding of coordinated rotational flight of pigeon flocks with infrequent directional switches.

However, before I can recommend publication in Royal Society Open Science, the authors should take care of the following issues.

► Thanks very much for your objective evaluation and valuable comments on our manuscript. The detailed responses are listed point by point below.

Comment 1. *There seems to be a large overlap with a previous paper of some of the authors [28]. In the Introduction and in the Results Section it should be clearly stated in which respects this paper differs from Ref.[28] Chen D, Xu B, Zhu T, Zhang HT. 2017 Anisotropic interaction rules in circular motions of pigeon flocks: An empirical study based on sparse bayesian learning.*

► Thanks for the constructive comment. As this reviewer noted, in Ref. [28], some authors of this manuscript, D. Chen and H. Zhang, have studied the circular motions of pigeon flocks. The previous published paper aims at revealing the interaction rules governing highly “coordinated free flight” of bird flocks. However, it adopted only a *linear* interaction assumption motivated by the standard Vicsek model via sparse Bayesian learning, since we have little prior knowledge about the interaction dynamics. Accordingly, it reported the observational results, including the number of interaction neighbors, the anisotropic interaction mechanism, and so on.

However, in the present manuscript, we focus on the role of nonlinear interactions in “coordinating directional switches” in pigeon flocks, which extends the usage of sparse Bayesian learning with a more general assumption of interaction strategy. More importantly, this work, for the first time, finds a mechanism that a certain degree of nonlinearity in interactions can enhance the coordinated rotational flight of pigeon flocks with infrequent directional switches. Such a mechanism is surprisingly contrary to the common knowledge: Linear and time-invariant interactions are sufficient to induce ordered collective motions.

Moreover, this work also establishes a data-driven particle model with two potential wells and estimates the mean switching times of rotational direction. Such a model with its analytical and numerical results further reinforces the mechanism that nonlinearity in interactions is truly beneficial to the emer-

gence of collective motion with less frequent switches.

Therefore, the two works focus on different research priorities in coordinated motion of pigeon flocks, and report different analytical, modeling, and experimental results. In order to clarify the difference between this paper and Ref.[28], we have added the following illustration to the section of Discussion in the revised manuscript.

“By using the SBL method, the interaction rules governing highly coordinated flight of pigeon flocks have been analyzed in our previous work [28]. However, the results in [28] are based on a linear interaction assumption motivated by the standard Vicsek model. Here, different from [28], we extended the usage of the SBL method and extracted the linearity and nonlinearity of the interaction from the smooth and the transient switching trajectories. We focus on the influence of nonlinear interaction on the directional switches of pigeon flocks.”

Comment 2. *All quantities should be defined when they first occur; e.g., on p.2 left, line 54, v_i (velocity, I assume) is not defined, and the formula for spin is not explained sufficiently (as angular momentum I would expect $r \times p$). It is not clear why this is a type of Vicsek model.*

► Thanks for the constructive comment. Indeed, v_i on page 2 means velocity, which has been defined in the revised version.

Moreover, it is important to understand that in the case of coordinated particles, a uniform rotation with the same rotation angle and a uniform rotation of the spin phase give two completely different results. The rotational movement in external space generated by the orbital angular momentum, acts on the positions of the points and therefore it gives rise to parallel paths trajectories: these all have the same origin as a positional centre of rotation, but different radii of curvature. On the other hand, the internal-space rotation phase generated by the spin s , acts on the velocities of the particles, giving rise to equal radius trajectories: these have different centres of rotation, but the same radius of curvature. Since spin plays a crucial role in rotational switches, we introduce the conception analogous to internal angular momentum in this work. In the revised manuscript, we have clearly show the definition of spin and one may find the details of the angular momentum from the previous study (J Stat. Phys. 158, 601–627, 2015, cited as ref. [44]) .

General Vicsek model describes the alignment rule of angles with neighboring particles. Here, we consider the spin interaction model in the Vicsek style, which means the interactions are mediated by the spins of the neighboring individuals.

Comment 3. *I assume that x in Eq. (4) and in Fig.2 is not the position coordinate of Fig.1a, as in Ref. [28], but the average spin S . Please clarify.*

► Thanks for the constructive comment. As the reviewer noted, x in Eq. (4) and in Fig.2 is not the position coordinate of Fig.1a. In fact, x in Eq. (4) means the independent variable in the interaction function. The function defines an interaction manner that individuals balance the trade-off between their spin direction and the alignment to their neighbors. We know that the rotation phase generated

by the spin s acts on the velocities, which exhibits a highly ordered state. Thus, we adopted the spin interaction model in this study.

Comment 4. *The nonlinear interaction function Eq.(4) is reminiscent of Arnold tongues in synchronization theory. Could the authors comment on that? The chosen value of alpha might then be compared to an effective coupling strength.*

► Thanks for the constructive comment. When some oscillatory processes are coupled, there exists the possibility of their becoming synchronized. Arnold proposed a map of the circle into itself, obtaining synchronization “tongues” and calculating their widths in the approximation of small coupling. In the theory of Arnold tongues (Akad. Nauk, Serie Math., 25, 21–96, 1961, cited as ref. [50] in the revised version), the strength of coupling has a strong influence on the locking region of synchronization. Analogously, the parameter α in our SDE model balances the linearity and nonlinearity of coupling strength, which influences the number of switches. Since the proposed model in this study is different from the Arnold’s oscillatory equations, the analytic tools are different.

Additionally, we have added the following sentences to the Discussion of the updated manuscript

“The SPP model is reminiscent of Arnold tongues in synchronization theory [50], where the strength of coupling has a strong influence on the locking region of synchronization. Analogously, the parameter α in our stochastic differential model balances the linearity and nonlinearity of coupling strength, which influences the number of switches.”

Comment 5. Minor remarks:

On p.2 right, line 30: give a reference for sparse Bayesian learning method

p.2 right, line 50: define i.i.d.

p.4 right, line 51, p.5 left, line 36 + 55 etc.: equilibria NOT equilibriums

p.5 left, line 23–25: sentence is not comprehensible

p.5 left, line 41: shown

Fig.2: The caption should better describe the role of alpha in the different red curves displayed. Give the range and the values of alpha.

Bibliography: give all names of coauthors.

error in author names in Ref. 29.

► Many thanks for your careful proofread. We have modified these errors and marked blue in the revised version.

Finally, we really appreciate all the useful and valuable suggestions from the editor and the reviewers. They are very constructive for improving the quality of this manuscript.

Now, we submit the revised manuscript to *Royal Society Open Science*. Thank you very much for editorially dealing our manuscript.

Sincerely,

Corresponding author: Wei Lin

Email: wlin@fudan.edu.cn